# Dynamic Planning of Mobile Service Teams' Mission Subject to Orders Uncertainty Constraints

**Grzegorz Bocewicz [1,2,\*]**, **Peter Nielsen [2]**, **Małgorzata Jasiulewicz-Kaczmarek [3]** and **Zbigniew Banaszak [1]**

[1] Faculty of Electronics and Computer Science, Koszalin University of Technology, 75-453 Koszalin, Poland; zbigniew.banaszak@tu.koszalin.pl

[2] Department of Materials and Production, Aalborg University, DK-9100 Aalborg, Denmark; peter@mp.aau.dk

[3] Faculty of Engineering Management, Poznan University of Technology, 60-965 Poznań, Poland; malgorzata.jasiulewicz-kaczmarek@put.poznan.pl

\* Correspondence: grzegorz.bocewicz@tu.koszalin.pl

**Abstract:** This paper considers the dynamic vehicle routing problem where a fleet of vehicles deals with periodic deliveries of goods or services to spatially dispersed customers over a given time horizon. Individual customers may only be served by predefined (dedicated) suppliers. Each vehicle follows a pre-planned separate route linking points defined by the customer location and service periods when ordered deliveries are carried out. Customer order specifications and their services time windows as well as vehicle travel times are dynamically recognized over time. The objective is to maximize a number of newly introduced or modified requests, being submitted dynamically throughout the assumed time horizon, but not compromising already considered orders. Therefore, the main question is whether a newly reported delivery request or currently modified/corrected one can be accepted or not. The considered problem arises, for example, in systems in which garbage collection or DHL parcel deliveries as well as preventive maintenance requests are scheduled and implemented according to a cyclically repeating sequence. It is formulated as a constraint satisfaction problem implementing the ordered fuzzy number formalism enabling to handle the fuzzy nature of variables through an algebraic approach. Computational results show that the proposed solution outperforms commonly used computer simulation methods.

**Keywords:** dynamic vehicle routing problem; ordered fuzzy numbers formalism; declarative modelling; service delivery management

## 1. Introduction

The Industry 4.0, also referred to as "smart" factory, and including solutions such as smart networking, mobility, flexibility of industrial operations and their interoperability, integration with customers and suppliers [1] using the possibilities of modern IT technologies, enables to monitor physical processes and make smart decisions through real-time communication and cooperation with humans, machines, sensors, etc. In this context, the Maintenance 4.0, also known as predictive maintenance, seems to be its main application area [2]. This is because by using advanced Artificial Intelligent (AI) methods to predict disruptions in the functioning of technical systems, predictive maintenance enables the minimization of downtime, prolonging machine life, increasing production efficiency, resource utilization, and reducing costs [3–5]. There is no common definition of Maintenance 4.0 or Industry 4.0, however, a number of studies undertaking these issues are growing rapidly and are also witnessed by many taxonomies of problems identified in both these areas, which presented, inter alia, in the works [2,6–8].

Technological changes such as the high need for transparency (e.g., supply chain visibility) and integrity control (right products, at the right time, place, quantity, condition, and at the right cost) in the supply chains make it possible to improve the level of requested services ordered by geographically dispersed customers. By analogy to the names of the aforementioned areas, the expectations mentioned here underlie the new concept of Logistics 4.0 [1].

In the context of the last two of the aforementioned concepts, i.e., Maintenance 4.0 and Logistics 4.0, it is worth paying attention to the next one called Perfective Maintenance. The idea behind this approach is to strive to improve the functioning of the system by supplementing it with additional functionalities and properties that improve it, e.g., improve accuracy, increase resistance, decrease to cost, etc. The essence of this concept, derived from Perfective Software Maintenance, the aim of which is to improve the performance (e.g., updating the software according to changes in the user interface), maintainability, or other attributes of a computer program [9]; it can also be used in systems providing ordered services with transport to the customer. The presented idea can be used in the course of maintenance of dispatcher's functionality responsible for planning of cyclically repeated delivery/service missions servicing spatially dispersed customers. In the considered case, the functioning of the vehicle fleet planning system could be improved by supplementing it with additional functionalities enabling to react to ad hoc changes in the ordered services. Consequently, such a perfective-maintenance-based approach concerned with the functional enhancement of the vehicle fleet planning system or enhancing its user interface would be especially useful in situations connected with the dynamic planning of milk-run driven systems providing ordered services while taking into account the constraints imposed by customer requests' uncertainty.

The milk-run routing and scheduling problems are usually recognized and formulated as a special case of the vehicle routing problem (VRP), [10–13]. Just as some authors distinguish between the inbound logistics referring to the transport, storage and delivery of goods coming into a business, and the outbound logistics referring to the same for goods going out of business [14], other authors distinguish in-plant milk-run (referring to raw materials, work in process and finished goods distribution) and out-plant milk-run supporting commodities and products transport between manufacturers and customers as well as service visits [15,16]. In both cases, the decisions regarding the vehicles routing policies are considered, i.e., the determination of routes along which customers are visited, and the schedule guaranteeing the congestion-free movement of the vehicles.

Milk-run problems usually concern planning routes that are cyclically repeated according to a fixed schedule in a fixed sequence and with fixed arrival times to plan whom to serve, how much to deliver and which regularly repeated routes to travel on using which fleet of vehicles. Relevant examples are provided by public transport systems including rail transport, urban transport, and intercity bus transport etc. Rhythmic delivery, repeated at regular intervals, is also a feature of systems of the cyclic delivery of food products to distribution centers, waste, recycling and composting pickup, packs to parcel locker-machine points, periodic service inspections as well as restocking beverages in street vending machines.

Since VRPs, which are Non-deterministic Polynomial-time (NP)-hard problems, only approximate solutions with the help of heuristic methods can be obtained [17,18]. In real-live cases, these kinds of problems become more complex due to the necessity of taking into account the influences caused by disruptions (following changes in execution of already planned deliveries and the appearance of new requests/orders, congestion or accidents) and the fuzzy nature of the parameters determining the timeliness of the performed services/deliveries. From a dynamic perspective, arising from the fact that orders are revealed incrementally over time, the considered outbound dynamic routing problem (DRP) consists of designing the vehicle routes (determined by customers' visit sequences) in an online fashion, i.e., communicating to the vehicle which customer to serve next as soon as the visit is accomplished. All related decisions are made without the knowledge of future orders. The need to take DRP commonly arises in the area of maintenance operations, where the ability to redirect a moving vehicle to a new request nearby allows for additional savings [19–21]. However, the fulfillment

of these expectations is conditioned by the ability to track the vehicle's position on an ongoing basis and communication ensuring the quick assignation of a new destination, i.e., with a guarantee of dynamically delivered services.

The uncertainty of DRP data due to traffic disruptions as well as changing the dates of the services completion implying the uncertainty of the final result force the necessity to adopt a model implementing the formalism of fuzzy sets. In turn, considering the necessity to take into account the aforementioned constraints of the nature of inequalities, implications and logical conditions, the declarative model seems to be best suited to guarantee these expectations. Therefore, the DRP can be formulated as a fuzzy constraint satisfaction problem and solved using both computer simulation and an analytical ordered-fuzzy-numbers-driven approach. It should be noted, that in opposition to standard fuzzy numbers, the support of the fuzzy number being a result of algebraic operations performed on ordered fuzzy numbers domain does not expand. This is the reason why the proposed use of the oriented fuzzy numbers algebra increases the competitiveness of the analytical approach in relation to the time-consuming computer simulation-based calculations of the feasible scenario of outbound mobile teams' dynamic rerouting.

In this context, the purpose of our research was to develop an ordered-fuzzy-numbers-driven declarative model, enabling to define the DRP subject to fuzzy maintenance time and transportation time constraints, the solution to which provides the possible dynamic rerouting scenarios. Unlike most of the problems discussed in the literature which focus on the search for solutions that optimize the path traveled or the cumulative cost of the mission carried out, in our approach, an answer to the following question was sought: can the newly reported delivery requests or the performance date correction of the already requested ones be accepted or not?

The present study is a continuation of our previous work that explored methods of the fast prototyping of solutions to the problems related to the routing and scheduling of tasks typically performed in batch flow production systems [22–28]. Its main contribution are threefold:

- Outbound mobile teams-driven maintenance services require taking into account disruptions occurring in road traffic (e.g., congestion-restricted delivery time) and the uncertainty of the delivery (e.g., unpacking and storage) operations or maintenance (e.g., repair or condition monitoring) services as well as changing the ordered dates of the service/delivery performance.
- Formulation of the DRP implementing the algebra of ordered fuzzy numbers allows one to plan mobile teams' operation, taking into account the uncertainty of their travel time and the time of conducted repairs.
- In opposition to standard fuzzy numbers, the support of the fuzzy number as a result of algebraic operations performed on ordered fuzzy numbers domain does not expand, which determines its dominance on the currently used computer simulation methods, the proposed algebraic approach allows for online vehicle rerouting and/or rescheduling forced by disturbances caused by ad hoc changes in the orders performed.

The structure of the paper is organized as follows. Section 2 includes the review of the literature. Section 3 provides preliminaries briefly referring to some known concepts from ordered fuzzy numbers theory and constraint programming techniques. The problem statement and the methodology used for its solution are described in Sections 4 and 5, respectively. Computational results are then reported and analyzed in Section 6, while conclusions and future directions of work are considered in Section 7.

## 2. Related Work

Most of the problems appearing in the milk-run systems are aimed at searching for an optimal periodic distribution policy. Examples of such problems [15,29] include both simple ones, e.g., Mix Fleet VRP, Multi-depot VRP, Split-up Delivery VRP, Pick-up and Delivery VRP, VRP with Time Windows, VRP with Backhauls, and more complex ones, e.g., VRP with multi-trip multi-traffic pick-up and delivery problem with time windows and synchronization being a combination of variants of the

vehicle routing problem with multiple trips, a vehicle routing problem with a time window, and a vehicle routing problem with pick-up delivery. Since milk-run routing and scheduling problems follow VRPs which are NP-hard, hence their solutions derived from the milk-run distribution policy while, for instance, aimed at determining in what time windows parts, can be collected from suppliers, and how many logistic trains and along which routes they should run, can be obtained with the help of heuristic methods [17,18,20,30]. Regardless of the class of the problems whether typical for in-plan or out-plant milk-run systems [14] or accentuating either the dynamic or static character of vehicle routing [15,17,21,29,30], their goal is to search for optimal solutions. These studies implicitly assume that there exist admissible solutions, e.g., ones that ensure the congestion-free flow of concurrently executed transport processes [31,32] and/or that planned routings and schedules are robust to assumed disruptions [20,21]. The most studies, which address outbound milk-run systems, focus on the routings and schedules of the vehicle fleet used. Most of the implemented mathematical model-based frameworks employ heuristic approaches using different metaheuristics, such as hybrid ant colony optimization and Tabu search.

It is worth noting that among the aforementioned issues, relatively few studies are devoted to the problems of outbound milk-run dynamic routing and the systems in which services are provided by appointment. In systems of this type, the dynamic multi-period vehicle problem is solved, which boils down to services scheduling being implemented in a rolling horizon fashion, in which new requests are received while unfulfilled during the first period together with the set of customer requests preplanned for the next period constitute the new portfolio of orders to be considered for subsequent scheduling [13,16,33]. Mentioned approaches do not take into account many the practical requirements and limitations imposed by, for example, the need to take into account the specificity of the same services and the capabilities of the teams performing these services. In general, in addition to the need to balance the needs of the serviced customers with the capability of the team implementing the ordered services, the issues of the synchronization of works carried out for a given user by various service teams (e.g., in mutual exclusion or rendez-vous mode) should also be noticed. A broad review of VRP taxonomy-inspired problems formulated in the milk-run systems class are presented in the works [10–12,19].

In many real situations, DRP data uncertainty due to traffic disruptions (uncertain travel times caused by weather conditions, daily changes in traffic intensity etc.) as well as the degree of difficulty of the service provided (caused by intertwined overhauls, condition monitoring, product repairs operations, etc.) cannot be valued in a precise way. However, the minority of models of the so-called Fuzzy VRP only assume vagueness for fuzzy demands to be collected and fuzzy service or travel times. Literature on these issues is very scarce [34], despite the rapidly growing demand for predictive maintenance-oriented service providers [10]. The rapidly developing enterprise servitization indicates the growing demand for this type of services [35–37].

It is worth adding that the development of the servitization-based approach is determined by the ability to reconfigure a delivery/service system, e.g., by taking into account the change of used vehicles' number and their capacity, the number and location of refilling stations (concerning fuel, tools, materials) and so on. In that context, the reconfigurability of the outbound milk-run driven delivery/services system can be seen as the answer to expectations related to achieving the desired level of system flexibility as well as the requirements of the outbound logistics resilience (referring to maintaining the assumed system's stability and robustness levels). It is worth noting that such challenges fit into the concept of intertwined supply network viability, guaranteeing survival in a changing environment [38].

To summarize, the presented review shows that there is an urgent need to develop analytical methods that would replace the labor-intensive and time-consuming methods of the computer simulation-based assessment of possible maintenance service scenarios. The methods sought should take into account the fact that the mobile service missions carried out require taking into account the uncertainty factor resulting from the fuzzy nature of the vehicle movement and services period.

It seems that the requirements mentioned above meet our approach, which combines the declarative modeling paradigm (implemented through the constraints of programming techniques) with an algebra of ordered fuzzy numbers.

## 3. Preliminaries

### 3.1. An Ordered Fuzzy Numbers Framework

The routing and scheduling problems developed to date have limited use due to the data uncertainty observed in practice. The values describing parameters such as transport time or loading/ unloading times depend on the human factor, which means they cannot be determined precisely. It is difficult to account for data uncertainty by using fuzzy variables due to the imperfections of the classical fuzzy numbers algebra [26]. Equations which describe the relationships between fuzzy variables (variables with fuzzy values) using algebraic operations (in particular, addition and multiplication) do not meet the conditions of the Ring (among others if the condition $\forall_{A \in \mathcal{F}} A + 0 = A$ is met, then condition $\forall_{A \in \mathcal{F}} \exists!_{B \in \mathcal{F}} A + B = 0$ is not met). In addition, algebraic operations based on standard fuzzy numbers follow Zadeh's extension principle. In practice, this means that no matter what algebraic operations are used, the support of the fuzzy number, which is the result of these operations, expands. Consequently, it is impossible to solve algebraic equations with fuzzy variables. In particular, this means that for any fuzzy numbers $a$, $b$, $c$, the following implication $(a + b = c) \Rightarrow [(c - b = a) \wedge (c - a = b)]$ does not hold. This makes it impossible to solve a simple equation $A + X = C$. This fact significantly hinders the use of approaches based on declarative models, in which most of the relationships between decision variables are described as linear/nonlinear equations and/or algebraic inequalities.

We address these issues by proposing the formalism of ordered fuzzy numbers (OFNs) algebra [39]:

**Definition 1.** *An OFN is a pair of continuous real functions:*

$$\hat{A} = (f_A,\ g_A),\ \text{where}:\ f_A,\ g_A \colon\ [0,1] \to \mathbb{R}. \tag{1}$$

The functions $f_A$ and $g_A$ are called the up part and the down part of the OFN $\hat{A}$, respectively. The values of these continuous functions are limited ranges, which can be defined as the following bounded intervals: $UP_A = (l_{A0}, l_{A1})$ and $DOWN_A = (p_{A1}, p_{A0})$. Assuming that $f_A$ is increasing and $g_A$ is decreasing as well as that $f_A \leq g_A$, the membership function $\mu_A$ of the OFN $\hat{A}$ is as shown in Figure 1a,b:

$$\mu_A(x) = \begin{cases} f_A^{-1}(x) & \text{when } x \in UP_A \\ g_A^{-1}(x) & \text{when } x \in DOWN_A \\ 1 & \text{when } x \in [l_{A1}, p_{A1}] \\ 0 & \text{otherwise} \end{cases} \tag{2}$$

A property called the orientation (direction) is defined for an OFN. There are two types of orientation: positive, when $\hat{A} = (f_A,\ g_A)$ the direction is consistent with the direction of the $OX$ axis, and negative, when $\hat{A} = (g_A,\ f_A)$, the direction is opposite to the direction of the $OX$ axis. Assuming that the values of all fuzzy variables may have a different orientation, the definitions of the algebraic operations used are as follows:

**Definition 2.** *Let $\hat{A} = (f_A,\ g_A)$ and $\hat{B} = (f_B,\ g_B)$ be OFNs. $\hat{A}$ is a number equal to $\hat{B}$ ($\hat{A} = \hat{B}$), $\hat{A}$ is a number greater than $\hat{B}$ or equal to or greater than $\hat{B}$ ($\hat{A} > \hat{B}$; $\hat{A} \geq \hat{B}$), $\hat{A}$ is less than $\hat{B}$ or equal to or less than $\hat{B}$ ($\hat{A} < \hat{B}$, $\hat{A} \leq \hat{B}$) if: $\forall_{x \in [0,1]} f_A(x) * f_B(x) \wedge g_A(x) * g_B(x)$, where the symbol $*$ stands for: $=, >, \geq, <, $ or $\leq$.*

**Definition 3.** *Let $\hat{A} = (f_A,\ g_A)$, $\hat{B} = (f_B,\ g_B)$, and $\hat{C} = (f_C,\ g_C)$ be OFNs. The operations of addition $\hat{C} = \hat{A} + \hat{B}$, subtraction $\hat{C} = \hat{A} - \hat{B}$, multiplication $\hat{C} = \hat{A} \times \hat{B}$ and division $\hat{C} = \hat{A}/\hat{B}$ are defined as follows: $\forall_{x \in [0,1]} f_C(x) = f_A(x) * f_B(x) \wedge g_C(x) = g_A(x) * g_B(x)$, where the symbol $*$ stands for $+, -, \times, $ or $\div$. The operation of division is defined for $\hat{B}$ such that $|f_B| > 0$ and $|g_B| > 0$ for $x \in [0,1]$.*

In recent years, the concept of OFNs has been continuously developed and used in various practical applications. Many publications have been devoted to the analysis of the OFN model in relation to convex fuzzy sets [40–43].

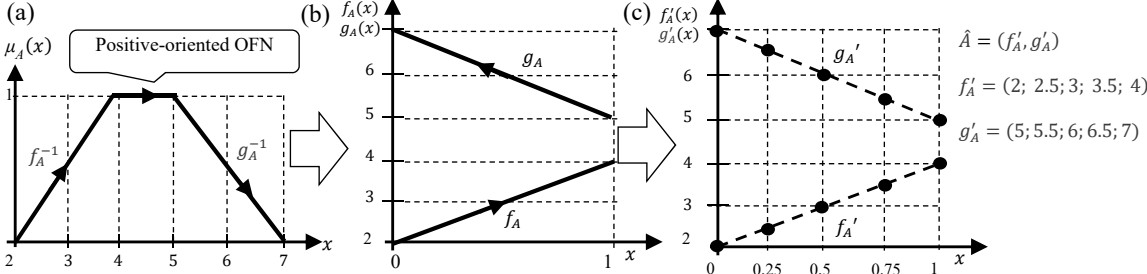

**Figure 1.** (**a**) Ordered fuzzy number (OFN) $\hat{A}$ represented as a convex fuzzy number; (**b**) functions $f_A$, $g_A$ determining $\hat{A}$ (positive orientation); and (**c**) the discrete representation of $\hat{A}$ ($dx = 0.5$).

### 3.2. Illustrative Example

Let us consider the graph $G = (N, E)$ modelling a transportation network composed of $|N| = \omega = 11$ delivery points (hereinafter referred to as nodes), i.e., customers and the service base, as shown in Figure 2. The points include 1 node representing the service point $N_1$ and 10 nodes representing customers $N_2$–$N_{11}$. The customers $N_2$–$N_{11}$ are cyclically serviced (with period $T = 2000$ u.t.) by the mobile service teams (MSTs) traveling form node $N_1$. The beginning moment of the node $N_\lambda$ occupation (service) by team $U_k$ is described by variable $y_\lambda^k$. The service is executed in intervals determined by the service deadline $\Delta_\lambda = [ld_\lambda; ud_\lambda] \in \Delta$ (see Table 1), i.e., $y_\lambda^k \geq ld_\lambda$ and $y_\lambda^k + t_\lambda \leq ud_\lambda$ (where $t_\lambda$ is time of node $N_\lambda$ occupation). Moreover, each node $N_\lambda$ can be serviced by MSTs offering required qualifications and confirmed with the appropriate certificates. The considered sets of qualifications $\psi_\lambda \in \Psi$ which are required by customers $N_\lambda$ are shown in Table 2.

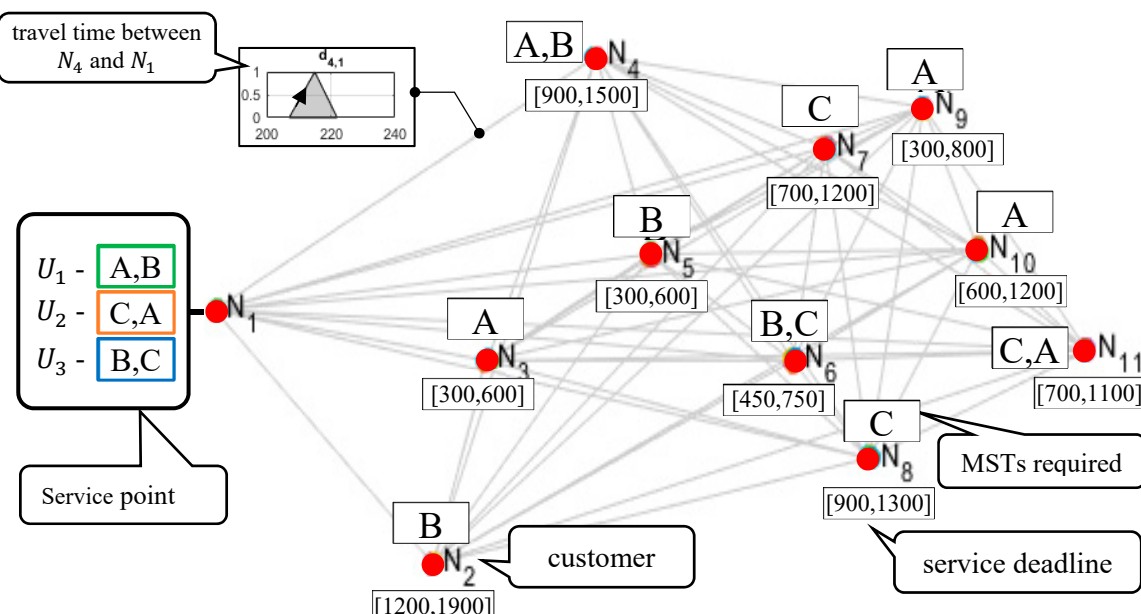

**Figure 2.** Graph $G$ modeling transportation network.

**Table 1.** Service deadlines for customers $N_2$–$N_{11}$.

|          | $N_2$ | $N_3$ | $N_4$ | $N_5$ | $N_6$ | $N_7$ | $N_8$ | $N_9$ | $N_{10}$ | $N_{11}$ |
|----------|-------|-------|-------|-------|-------|-------|-------|-------|----------|----------|
| $ld_\lambda$ | 1200 | 300 | 900 | 300 | 450 | 700 | 900 | 300 | 600 | 700 |
| $ud_\lambda$ | 1900 | 600 | 1500 | 600 | 750 | 1200 | 1300 | 800 | 1200 | 1100 |

**Table 2.** Sets of required qualifications $N_2$–$N_{11}$.

|          | $N_2$ | $N_3$ | $N_4$ | $N_5$ | $N_6$ | $N_7$ | $N_8$ | $N_9$ | $N_{10}$ | $N_{11}$ |
|----------|-------|-------|-------|-------|-------|-------|-------|-------|----------|----------|
| $\psi_\lambda$ | {B} | {A} | {A, B} | {B} | {B, C} | {C} | {C} | {A} | {A} | {C, A} |

For example, customer $N_4$ should be serviced within the interval time [900; 1500] by MSTs offering qualifications $A$ and $B$ (one MST offering set {$A, B$} or two MSTs: the first offering $A$ and the second offering $B$).

Each edge $\left(N_\beta, N_\lambda\right) \in E$ linking nodes $N_\beta$ and $N_\lambda$ is labelled with a fuzzy variable (in the OFN representation) representing the uncertainty of the traveling time $d_{\beta,\lambda}$ between the nodes $N_\beta$ and $N_\lambda$ (see Figure 3). Given is a set of MSTs $\mathcal{U} = \{U_1, \ldots, U_k, \ldots, U_K\}$ servicing customers spatially dispersed in network $G$.

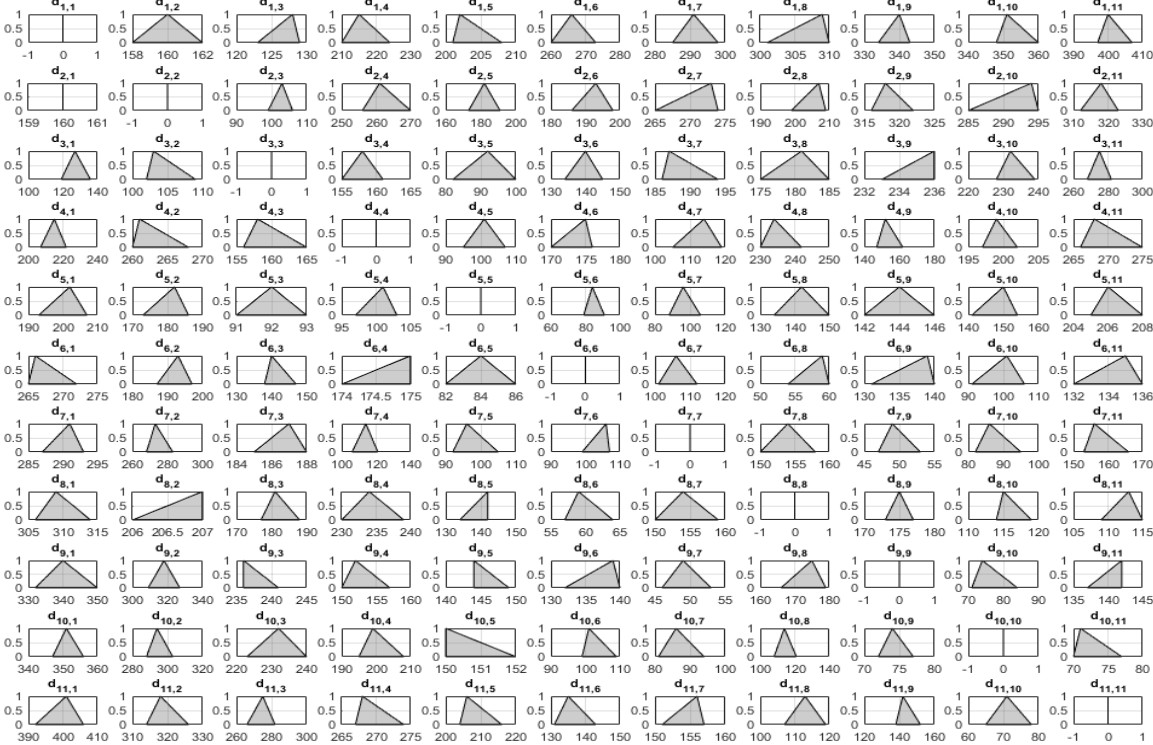

**Figure 3.** Traveling times between nodes for the networks from Figure 2 (an OFN representation).

For each $U_k$ the set $\Phi_k$ of the offered qualifications is assigned. For example, the available set $\mathcal{U} = \{U_1, U_2, U_3\}$ in a network $G$ (Figure 2) contains three MSTs offering the following qualifications: $\Phi_1 = \{A, B\}$; $\Phi_2 = \{C, A\}$; $\Phi_3 = \{B, C\}$. This means that:

- The team $U_1$ can completely satisfy the expectations of the nodes: $N_2$, $N_3$, $N_4$, $N_5$, $N_9$, $N_{10}$, and partially those of the nodes: $N_6$, $N_{11}$;
- The team $U_2$ can completely satisfy the expectation of nodes: $N_3$, $N_7$, $N_8$, $N_9$, $N_{10}$, $N_{11}$, and partially of nodes: $N_4$, $N_6$;

- The team $U_3$ can completely satisfy the expectations of the nodes: $N_2$, $N_5$, $N_6$, $N_7$, $N_8$, and partially those of the nodes: $N_4$, $N_5$ and $N_{11}$.

The routes traveled by team $U_k$ are denoted by sequences of nodes: $\pi_k = (N_{k_1}, \ldots, N_{k_i}, N_{k_{i+1}}, \ldots, N_{k_\mu})$, where $k_i \in \{1, \ldots, K\}$, $\forall_{k_i \neq k_j} N_{k_i} \neq N_{k_j}$, $(N_{k_i}, N_{k_{i+1}}) \in E$. Nodes representing the service point (e.g., $N_1$) appear along every route. Moreover, each route $\pi_k$ consists of nodes in which customers $N_\lambda$ assigned to them expect services that require qualifications $\psi_\lambda$, i.e., for each team $U_k$ offering qualifications $\Phi_k$ the following condition holds $\Phi_k \cap \psi_\lambda \neq \varnothing$.

In this context, the problem of the proactive planning of service team trips boils down to the question: do the schedule and routings of MSTs guarantee the timely execution of the ordered services?

Given a set $\mathcal{U}$ of MSTs providing services (according to given qualifications $\Phi_k$) to the customers allocated in a network $G$ (ordering an assumed kind of services $\Psi$). Does there exist a set of routes $\Pi$ guaranteeing the timely execution of the ordered services (according to given service deadlines $\Delta_\lambda$)?

The examples of such routes $\Pi$ and the associated fuzzy schedule for the network $G$ for Figure 2 are illustrated in Figures 4 and 5. The routes are specified by the sequences of nodes: $\pi_1 = (N_1, N_9, N_{10}, N_4, N_1)$, $\pi_2 = (N_1, N_3, N_{11}, N_1)$, $\pi_3 = (N_1, N_5, N_6, N_7, N_8, N_2, N_1)$. It should be noted that in the presented solution, customer service is provided only by the necessary MSTs. Moreover, despite the uncertain (fuzzy) traveling times $d_{\beta,\lambda}$, it is also assumed that all customers are serviced cyclically (with period $T = 2000$) due to given service deadlines $\Delta_\lambda$—see Figure 5.

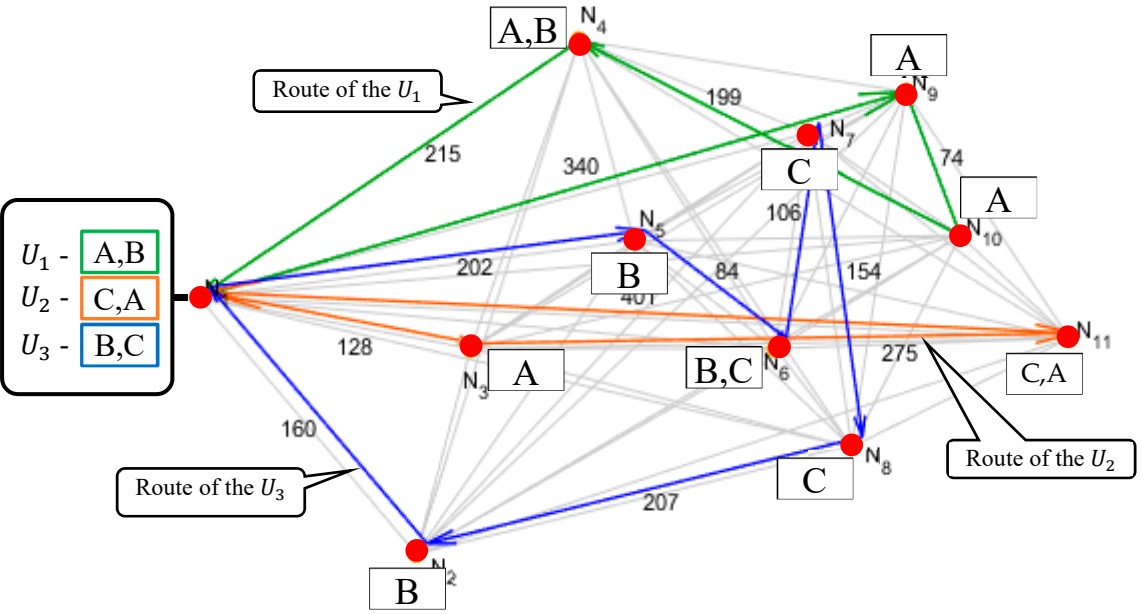

**Figure 4.** The routes $\Pi$ of $\mathcal{U} = \{U_1, U_2, U_3\}$ guaranteeing the timely service of the customers $N_2 - N_{11}$.

Due to the occurrence of unforeseen disturbances, the implementation of proactively designated customers service plans becomes practically impossible. An example of such a disturbance are the unforeseen changes of service deadlines. Such a kind of disturbance is presented in Figure 5 where the dispatcher receives information about changing the date of the customer service being located at the node $N_6$ (from $\Delta_6 = [450; 750]$ to $\Delta_6^* = [650; 950]$), see the second window (moment $t^* = 2500$ when $U_1$ occupies $N_9$, $U_2$ occupies $N_3$ and $U_5$ occupies $N_5$). Due to this change, the adopted routes do not guarantee the implementation of maintenance services on the set dates—the handling of $N_6$ according to the new service deadline $\Delta_6^* = [650; 950]$ prevents the timely handling of the client $N_8$ and vice versa. In such a situation, it becomes necessary to answer the following question:

*Given a set $\mathcal{U}$ of MSTs providing services (according to given qualifications $\Phi_k$) to the customers allocated in a network (ordering assumed kind of services $\Psi$), MSTs move along a given set of routes $\Pi$ according to a cyclic fuzzy schedule $\hat{\mathbb{Y}}$. Given is a disturbance changing $\Delta_\lambda$ to $\Delta_\lambda^*$ at the moment $t^*$. Does there exist a rerouting $^*\Pi$ and rescheduling $\widehat{^*\mathbb{Y}}$: of MSTs, which guarantee the timely execution of the ordered services?*

The possibility of the reactive (dynamic) planning of MST missions in the event of the disruption of service deadlines is the subject of the following chapters.

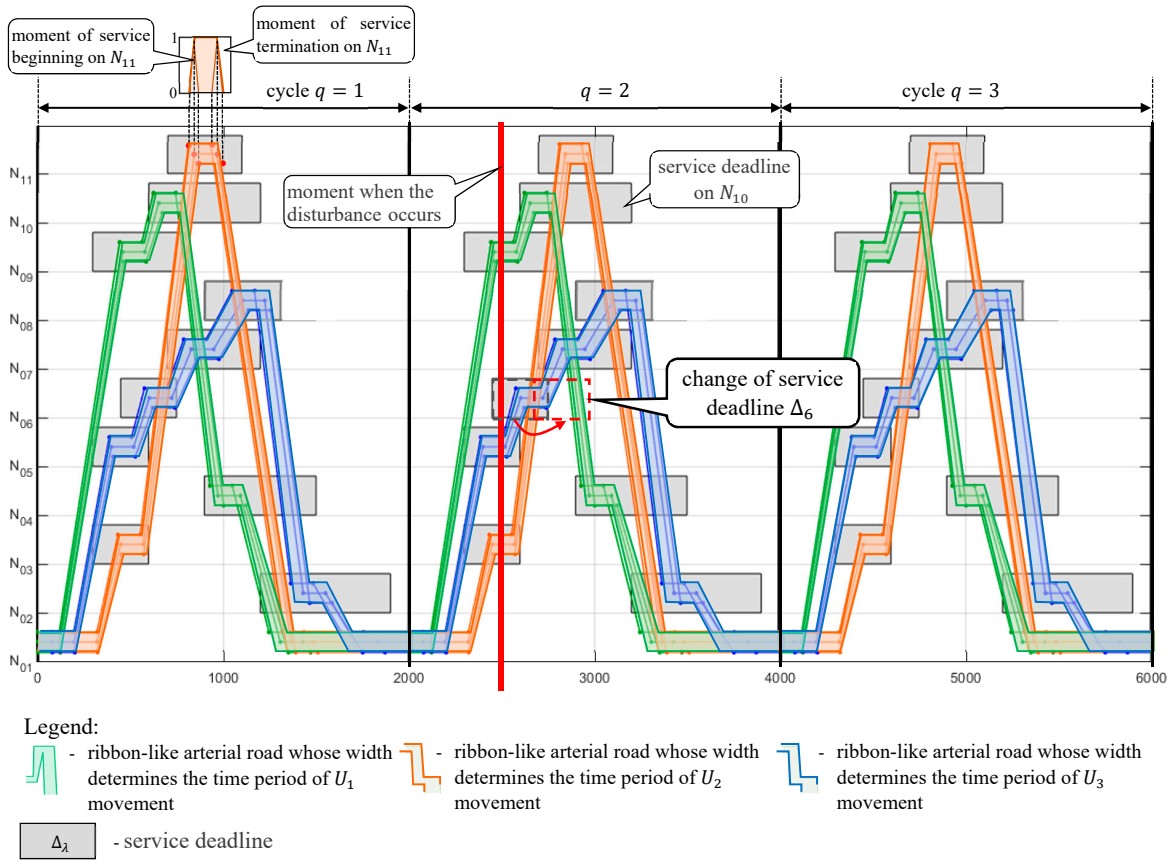

**Figure 5.** Fuzzy schedule for the implementation of maintenance services.

## 4. Problem Description

### 4.1. Assumptions

The following assumptions are met:

- Given is a network $G = (N, E)$,
- Each node $N_\lambda \in N$ is labelled with a fuzzy value $\widehat{t_\lambda}$ (represented in terms of OFN) denoting the duration of node occupation (service);
- Each edge $(N_\beta, N_\lambda) \in E$ is labeled with a fuzzy value $\widehat{d_{\beta,\lambda}}$ denoting the travel time between nodes $N_\beta$ and $N_\lambda$;
- Given is a set $\mathcal{U}$ of MSTs, in which each MST $U_k$ travels route $\pi_k$ ($\pi_k \in \Pi$);
- Each $U_k$ offers the set of qualifications $\Phi_k$ expected by the served customers;
- Node $N_1$ representing a service point occurs uniquely in all routes;
- Customer assigned to the node $N_\lambda$ ($\lambda > 1$) expects services that require proper set of qualifications $\psi_\lambda$;

- Each route $\pi_k$ consists of nodes in which customers $N_\lambda$ assigned to them expect services that require qualifications $\psi_\lambda$ following $\Phi_k \cap \ \psi_\lambda \neq \varnothing$;
- Customers are serviced cyclically in time windows repeated with period $T$;
- Customer assigned to the node $N_\lambda$ is serviced by deadline $\Delta_\lambda = [ld_\lambda; ud_\lambda]$;
- Fuzzy beginning moments $\widehat{y}_\lambda^k$ (represented as OFN) of the node $N_\lambda$ occupation make up final fuzzy cyclic schedule $\widehat{\mathbb{Y}}$;
- Disruption is understood as a change of the service deadlines from $\Delta$ to $\Delta^*$;
- Moment $t^*$ determines the disruption occurring.

It is also assumed that—values of the decision variables are represented as OFN, see Definition 1. Consequently, the OFN $\hat{A}$ is described by sequences $f_A'$ and $g_A'$ containing the values of functions $f_A$ and $g_A$ obtained by discretization of the interval $[0, 1]$:

$$f_A' = (f_A(0),\ f_A(dx),\ \ldots,\ f_A((M-1)\ dx),\ f_A(1)), \tag{3}$$

$$g_A' = (g_A(1),\ g_A((M-1)dx),\ \ldots,\ g_A(1dx),\ g_A(0)),\ dx = \frac{1}{M}, \tag{4}$$

where $(M + 1)$ is the number of samples (Figure 1c).

### 4.2. Declarative Model

Following the assumptions stated above, the proposed reference model consists of:

Parameters:

Crisp parameters:

$G$ : graph of a transportation network $G = (N, E)$, where $N = \{N_1, \ldots, N_\lambda, \ldots, N_n\}$ is a set of nodes and $E = \left\{ \left(N_i, N_j\right) \middle| i,\ j \in N, i \neq j \right\}$ is a set of edges, $n$—the number of nodes;

$\mathcal{U}$: set of MSTs: $\mathcal{U} = \{U_1, \ldots, U_k, \ldots, U_K\}$, where $U_k$ is the $k$-th MST;

$K$: size of the fleet;

$\Psi$: family of required sets of service qualifications: $\Psi = \{\psi_1, \ldots, \psi_\lambda, \ldots, \psi_n\}$, where $\psi_\lambda$ is a set of qualifications required by customer $N_\lambda$ (see example in Figure 3);

$\Phi$: family of sets of offered qualifications: $\Phi = \{\Phi_1, \ldots, \Phi_k, \ldots, \Phi_K\}$, where $\Phi_k$ is a set of qualifications offered by $U_k$ (see example in Figure 3);

$\Delta$: set of service deadlines: $\Delta = \{\Delta_1, \ldots, \Delta_\lambda, \ldots, \Delta_n\}$, where $\Delta_\lambda = [ld_\lambda; ud_\lambda]$ is a deadline for service at the customer $N_\lambda$ (see example in Figure 5);

$IS$: disturbance $IS = (M, \Delta^*)$ where: $M$ is a state of fleet mission at the moment $t^*$: $M = ((\mu_1, \ldots, \mu_k, \ldots, \mu_K), t^*)$, where $\mu_k \in N$ is the node occupied by $U_k$ (or the node the $U_k$ is headed to) at time $t^*$, the information about the disturbance is received. For example, in the situation shown in Figure 5, the information about the disturbance $IS$ was received at moment $t^* = 2500$ where the mission state is equal to: $M = ((N_9, N_3, N_5), 2500)$;

$\Delta^*$: is a set of changed service deadlines (caused by the appearance of disturbances): $\Delta^* = \left\{ \Delta_1^*, \ldots, \Delta_\lambda^*, \ldots, \Delta_n^* \right\}$, where $\Delta_\lambda^* = \left[ ld_\lambda^*;\ ud_\lambda^* \right]$ is a new deadline (after the occurrence of disturbance) for providing a service to customer $N_\lambda$;

$T$: window width, understood as a period, repeated at regular intervals, in which all nodes should be serviced (see Figure 5–$T = 2000$);

$\Pi$: set of routes $\pi_k$ before the occurrence of the disturbance $IS$, where $\pi_k$ is a route of $U_k$:

$$\pi_k = \left(N_{k_1}, \ldots, N_{k_i}, N_{k_{i+1}}, \ldots, N_{k_\mu}\right),\ \text{where } x_{k_i, k_{i+1}}^k = 1 \text{ for } i = 1, \ldots,\ \mu - 1 \text{ and } x_{k_\mu, k_1}^k = 1$$

$$x^k_{\beta,\lambda} = \begin{cases} 1 & \text{if } U_k \text{ travels from node } N_\beta \text{ to node } N_\lambda \\ 0 & \text{otherwise} \end{cases}$$

Imprecise parameters: (defined as positive-oriented OFNs and marked by "^"):

$\widehat{d_{\beta,\lambda}}$: traveling time along edge $(N_\beta, N_\lambda)$;

$\widehat{t_\lambda}$: time of node $N_\lambda$ occupation;

$\widehat{\mathbb{Y}}$: fuzzy schedule of fleet $\mathcal{U}$, $\widehat{\mathbb{Y}} = (\widehat{Y}, \widehat{W})$ before the disturbance *IS*:

$\quad \widehat{Y}$: family of $\widehat{Y^k}$, where $\widehat{Y^k}$ is a sequence of moments $\widehat{y^k_\lambda}$: $\widehat{Y^k} = (\widehat{y^k_1}, \ldots, \widehat{y^k_\lambda}, \ldots, \widehat{y^k_n})$, $\widehat{y^k_\lambda}$ is fuzzy time at which $U_k$ arrives at node $N_\lambda$;

$\quad \widehat{W}$: family of $\widehat{W^k}$, where $\widehat{W^k}$ is a sequence of laytimes $\widehat{w^k_\lambda}$: $\widehat{W^k} = (\widehat{w^k_1}, \ldots, \widehat{w^k_\lambda}, \ldots, \widehat{w^k_n})$, $\widehat{w^k_\lambda}$ is laytime at node $N_\lambda$ for $U_k$.

Variables:

Crisp variables:

$^*x^k_{\beta,\lambda}$: binary variable indicating the travel of $U_k$ between nodes $N_\beta$, $N_\lambda$ after disturbance *IS*:

$$^*x^k_{\beta,\lambda} = \begin{cases} 1 & \text{if } U_k \text{ travels from node } N_\beta \text{ to node } N_\lambda \\ 0 & \text{otherwise} \end{cases}$$

Imprecise variables (positive-/negative-oriented OFNs):

$^*\widehat{y^k_\lambda}$: fuzzy time at which $U_k$ arrives at node $N_\lambda$, after occurrence of the disturbance *IS*;

$^*\widehat{w^k_\lambda}$: laytime at node $N_\lambda$ for $U_k$, after occurrence of the disturbance *IS*;

$^*\widehat{s^k}$: take-off time of $U_k$.

Sets and sequences:

$^*\pi_k$: route of $U_k$, after occurrence of the disturbance *IS*: $^*\pi_k = (N_{k_1}, \ldots, N_{k_i}, N_{k_{i+1}}, \ldots, N_{k_\mu})$, where:

$$^*x^k_{k_i,k_{i+1}} = 1 \text{ for } i = 1, \ldots, \mu - 1 \text{ and } ^*x^k_{k_\mu,k_1} = 1;$$

$^*\Pi$: set of routes $^*\pi_k$;

$^*\widehat{W^k}$: sequence of laytimes $^*\widehat{w^k_\lambda}$: $^*\widehat{W^k} = (^*\widehat{w^k_1}, \ldots, ^*\widehat{w^k_\lambda}, \ldots, ^*\widehat{w^k_n})$;

$^*\widehat{W}$: family of $^*\widehat{W^k}$;

$^*\widehat{Y^k}$: sequence of moments $^*\widehat{y^k_\lambda}$: $^*\widehat{Y^k} = (^*\widehat{y^k_1}, \ldots, ^*\widehat{y^k_\lambda}, \ldots, ^*\widehat{y^k_n})$;

$^*\widehat{Y}$: family of $^*\widehat{Y^k}$;

$^*\widehat{\mathbb{Y}}$: fuzzy schedule of fleet $\mathcal{U}$, after occurrence of the disturbance *IS*: $^*\widehat{\mathbb{Y}} = (^*\widehat{Y}, ^*\widehat{W})$.

Constraints:

Routes. Relationships between the variables describing MST take-off times/mission start times and the task order:

$$^*\widehat{s^k} \geq 0 \, ; \, k = 1 \ldots K, \tag{5}$$

$$\left( \widehat{s^k} \leq t^* \right) \Rightarrow \left( ^*\widehat{s^k} = \widehat{s^k} \right); \, k = 1 \ldots K \tag{6}$$

$$\left(\widehat{y_j^k} \leq t^*\right) \Rightarrow \left({}^*x_{i,j}^k = x_{i,j}^k\right); \ j = 1 \ldots n; \ i = 2 \ldots n; \ k = 1 \ldots K, \tag{7}$$

$$\left(\widehat{y_j^k} \leq t^*\right) \Rightarrow \left(\widehat{{}^*y_j^k} = \widehat{y_j^k}\right); \ j = 2 \ldots n; k = 1 \ldots K, \tag{8}$$

$$\left(\widehat{y_j^k} \leq t^*\right) \Rightarrow \left(\widehat{{}^*w_j^k} = \widehat{w_j^k}\right); \ j = 2 \ldots n; k = 1 \ldots K, \tag{9}$$

$$\sum_{j=1}^{n} {}^*x_{1,j}^k = 1 \ ; \ k = 1 \ldots K, \tag{10}$$

$$\left({}^*x_{1,j}^k = 1\right) \Rightarrow \left(\widehat{{}^*y_j^k} = \widehat{{}^*s^k} + \widehat{d_{1,j}}\right); \ j = 1 \ldots n; \ k = 1 \ldots K, \tag{11}$$

$$\left(\widehat{{}^*y_j^k} > 0 \ \wedge \widehat{{}^*y_j^q} > 0\right) \Rightarrow \left(\left|\widehat{{}^*y_j^k} - \widehat{{}^*y_j^q}\right| \geq 0\right); i = 1 \ldots n; \ k, q = 1 \ldots K; \ k \neq q, \tag{12}$$

$$\left({}^*x_{i,j}^k = 1\right) \Rightarrow \left(\widehat{{}^*y_j^k} = \widehat{{}^*y_i^k} + \widehat{d_{i,j}} + \widehat{t_i} + \widehat{{}^*w_i^k}\right); \ j = 1 \ldots n; \ i = 2 \ldots n; \ k = 1 \ldots K, \tag{13}$$

$$\left(\Phi_k \cap \psi_j = \varnothing \right) \Rightarrow (\sum_{i=1}^{n} {}^*x_{i,j}^k = 0), \ j = 2 \ldots n; k = 1 \ldots K, \tag{14}$$

$$\cup_{k \in \mathcal{X}_j} \Phi_k = \psi_j, \ j = 2 \ldots n, \ \mathcal{X}_j = \{k : \ \sum_{i=1}^{n} {}^*x_{i,j}^k > 0\} \tag{15}$$

$$\widehat{{}^*s^k} + T = \widehat{{}^*y_1^k} + \widehat{t_1} + \widehat{{}^*w_1^k}; \ k = 1 \ldots K, \tag{16}$$

$$\widehat{{}^*y_j^k} \geq 0; \ i = 1 \ldots n; \ k = 1 \ldots K, \tag{17}$$

$$\sum_{j=1}^{n} {}^*x_{i,j}^k = \ \sum_{j=1}^{n} {}^*x_{j,i}^k; \ i = 1 \ldots n; \ k = 1 \ldots K, \tag{18}$$

$$\widehat{{}^*y_i^k} \leq T, \ i = 1 \ldots n; \ k = 1 \ldots K, \tag{19}$$

$${}^*x_{i,i}^k = 0; \ i = 1 \ldots n; \ k = 1 \ldots K. \tag{20}$$

Service deadlines. All customers $N_\lambda$ should be serviced by the given deadlines $\Delta_\lambda^* = \left[ld_\lambda^*; \ ud_\lambda^*\right]$:

$$\widehat{{}^*y_i^k} + \widehat{t_i} + c \times T \ \leq \ ud_\lambda^*, \ i = 1 \ldots n; \ k = 1 \ldots K, \tag{21}$$

$$\widehat{{}^*y_i^k} + c \times T \ \geq \ ld_\lambda^*, \ i = 1 \ldots n; \ k = 1 \ldots K. \tag{22}$$

### 4.3. Fuzzy Constraint Satisfaction Problem

The model proposed above allows to define the problem under consideration in the following way:

*Given a set $\mathcal{U}$ of MSTs servicing customers allocated in a network G (customers are serviced by prescheduled deadlines $\Delta$), MSTs move along a given set of routes $\Pi$ according to a cyclic fuzzy schedule $\widehat{\mathbb{Y}}$. Assuming that there occurs a disturbance IS which changes $\Delta$ to $\Delta^*$, a feasible way of rerouting ($^*\Pi$) and rescheduling ($^*\widehat{\mathbb{Y}}$) of MSTs, guaranteeing timely execution of the ordered services, is sought.*

The response to the signaled disturbance *IS* is the rescheduling and rerouting of the MSTs resulting then in a new plan of service delivery. In that context, when disturbance *IS* occurs, the new set of routes $^*\Pi$ and a new schedule $^*\widehat{\mathbb{Y}}$, which guarantees the timely servicing of customers, are determined by solving the following fuzzy constraint satisfaction (FCS) problem (23):

$$\widehat{FCS}\left(\widehat{\mathbb{Y}}, \Pi, \ IS\right) = \left(\left(\widehat{\mathcal{V}}, \widehat{\mathcal{D}}\right), \widehat{C}\left(\widehat{\mathbb{Y}}, \Pi, \ IS\right)\right), \tag{23}$$

where:

$\widehat{\mathcal{V}} = \left\{ {}^*\widehat{\mathbb{Y}}, {}^*\Pi \right\}$ is a set of decision variables: ${}^*\widehat{\mathbb{Y}}$—a fuzzy cyclic schedule guaranteeing the timely provision of service to customers in the case of disturbance *IS*, and ${}^*\Pi$—a set of routes determining the fuzzy schedule ${}^*\widehat{\mathbb{Y}}$;

$\widehat{\mathcal{D}}$— a finite set of decision variable domains: ${}^*\widehat{y^k_\lambda}, {}^*\widehat{w^k_\lambda} \in \mathcal{F}$ ($\mathcal{F}$ is a set of OFNs (1)), ${}^*x^k_{\beta,\lambda} \in \{0, 1\}$;

$\widehat{\mathcal{C}}$— a set of constraints which take into account the set of routes $\Pi$, fuzzy schedule $\widehat{\mathbb{Y}}$ and disturbance *IS*, while determining the relationships that link the operations occurring in MSTs cycles (5)–(22).

To solve $\widehat{FCS}$ (23), it is necessary to determine the values of the decision variables from the adopted set of domains for which the given constraints are satisfied. The implementation of $\widehat{FCS}$ in a constraint programming environment, such as IBM CPLEX ILOG, enables to find the solution.

## 5. Solution Methodology

The approach proposed assumes that the reaction to randomly occurring disruptions *IS* (resulting in, e.g., resignation from services and/or change of the dates of their implementation) takes place on an ongoing basis in the online mode. This is done through dynamic adaptation (i.e., the rerouting and rescheduling) of previously adopted routes $\Pi$, and schedules $\widehat{\mathbb{Y}}$, i.e., adjusting them (if possible) to the changes in services timetable.

It is understood that the considered output schedule $\widehat{\mathbb{Y}}$ sets the dates of periodically performed inspections/service repairs ordered by customers. Let $\widehat{\mathbb{Y}}(q)$ denote the fuzzy schedule of the *q*-th cycle defined as

$$\widehat{\mathbb{Y}}(q) = \left( {}^*\widehat{Y}(q), {}^*\widehat{W}(q) \right) \tag{24}$$

where ${}^*\widehat{Y}(q)$ and ${}^*\widehat{W}(q)$ are families of the following sets:

$$\widehat{{}^*Y^k}(q) = \left( \widehat{{}^*y^k_1}(q), \dots, \widehat{{}^*y^k_\lambda}(q), \dots, \widehat{{}^*y^k_n}(q) \right) \text{ and } \widehat{{}^*y^k_\lambda}(q) = \widehat{{}^*y^k_\lambda} + (q-1) \times T, \ q = 1, 2 \dots, Q$$

$$\widehat{{}^*W^k}(q) = \left( \widehat{{}^*w^k_1}(q), \dots, \widehat{{}^*w^k_\lambda}(q), \dots, \widehat{{}^*w^k_n}(q) \right) \text{ and } \widehat{{}^*w^k_\lambda}(q) = \widehat{{}^*w^k_\lambda} + (q-1) \times T, \ q = 1, 2 \dots, Q$$

The considered implementations of recurring service missions describe the routes $\Pi$ and schedules: $\widehat{\mathbb{Y}}(1), \widehat{\mathbb{Y}}(2), \dots, \widehat{\mathbb{Y}}(Q)$ sequences, where $Q$ is the number of cycles performed. It is assumed that disturbance *IS* can occur in any cycle $q$.

An algorithm that supports dynamic planning, i.e., vehicle fleet rerouting and rescheduling, based on the proposed concept of $\widehat{FCS}$ (23), is shown in Figure 6. The algorithm processes the successive customer service cycles $q = 1, 2 \dots, Q$. If there is a disturbance ($IS \neq \varnothing$) in a given cycle $q$ (at moment $t^*$), then the problem $\widehat{FCS}$ is solved (*solve* function). The function *solve* represents algorithms implemented in declarative programming environments (responsible for the search for admissible solutions to the decision problems considered).

The existence of an admissible solution (i.e., $\left( {}^*\widehat{\mathbb{Y}} \neq \varnothing \right) \wedge ({}^*\Pi \neq \varnothing)$) means that there are routes which ensure that customers are serviced on time when the disturbance *IS* occurs in the cycle $Q$. If an admissible solution does not exist, then the currently used routes and the associated vehicle schedule should be modified (*reduce* function) in such a way as to remove the servicing operation at node $N_\lambda$ at which disturbance *IS* occurs. The *reduce* function is responsible for modifying (rerouting) the routes. The proposed algorithm formulated in the constraints programming framework was implemented in the IBM CPLEX ILOG environment.

The presented algorithm generates in reactive mode (in situations of occurrence of service date change *IS*) alternative corrected versions of the assumed customer service plan. It needs to be highlighted that the proposed changes must not disrupt the timing of the customers' services to whom the disturbance does not apply to. Thus, there are situations in which such changes resulting in corrected versions of services delivery mission are not possible. In such cases, it is assumed that the affected customers will not be served in a given cycle (unhandled requests are not carried over to subsequent cycles).

The computational complexity of the algorithm from Figure 6 depends on the methods used to solve the problem $\widehat{FCS}$ (function *solve*). Due to the fact that the problem $\widehat{FCS}$ is an NP-hard possibility of the reactive change of assumed proactively scheduled services is limited to a small scale of problems. The assessment of the effectiveness of the proposed approach is the subject of the experiments described in the next section.

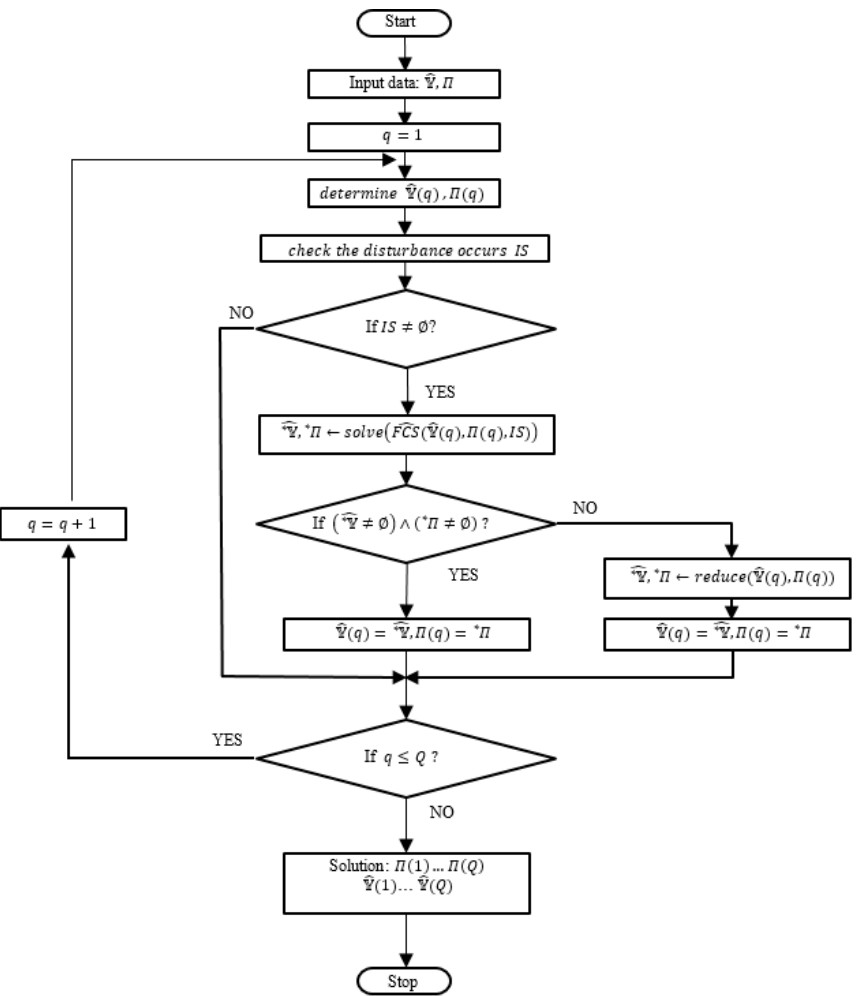

**Figure 6.** A dynamic rerouting and rescheduling algorithm.

## 6. Computational Experiments

Considering the graph model of the transportation network from Figure 2, in which three MSTs $\mathcal{U} = \{U_1, U_2, U_3\}$ periodically (with the period $T = 2000$ [u.t].) review the serviced stands owned, by using the customers located at nodes $N_2$–$N_{11}$, MSTs offer the following sets of qualifications: $\Phi_1 = \{A, B\}$; $\Phi_2 = \{C, A\}$; $\Phi_3 = \{B, C\}$. The assumed service deadlines $\Delta$, required qualifications $\Psi$ and fuzzy traveling times between the nodes $\widehat{d_{\lambda,\beta}}$ are collected in Tables 1 and 2, Figure 3, respectively. Routes $\pi_1 = (N_1, N_9, N_{10}, N_4, N_1)$, $\pi_2 = (N_1, N_3, N_{11}, N_1)$, $\pi_3 = (N_1, N_5, N_6, N_7, N_8, N_2, N_1)$ determine the fuzzy schedule $\widehat{\mathbb{Y}}$ of the service mission being carried out as shown in Figure 4. It is easy to see (Figure 4) that in the second cycle of the fuzzy schedule (in the state $M = ((N_9, N_3, N_5), 2500)$), an information about suddenly reported changes in the service deadline $\Delta_6^* = [450; 750]$ (instead $\Delta_6 = [650; 950]$) on node $N_6$ is announced. Given this, an answer to the following question is sought:

> *Does there exist a set of routes $^*\Pi$ operated by MSTs $U_1$, $U_2$ and $U_3$ for which the fuzzy cyclic schedule $^*\widehat{\mathbb{Y}}$ will guarantee that all customers are serviced on time when disturbance $IS = (S, \Delta^*)$ occurs?*

In order to find the answer to this question, the algorithm shown in Figure 6 has been employed. The problem $\widehat{FCS}$ (23) was then implemented in IBM ILOG CPLEX (Windows 10, Intel Core Duo2 3.00 GHz, 4 GB RAM).

The solution time for the problems of this size does not exceed 60 s—see Figure 7c. The following routes were obtained: $^*\pi_1 = (N_1, N_9, N_6, N_{10}, N_4, N_1)$, $^*\pi_2 = (N_1, N_3, N_6, N_7, N_1)$, $^*\pi_3 = (N_1, N_5, N_{11}, N_6, N_8, N_2, N_1)$. It should be noted that the new routes provide simultaneous customer service $N_6$ by two MSTs: $U_1, U_2$ (whose qualifications meet the required service needs: $\psi_6 \subset \Phi_1 \cup \Phi_2$).

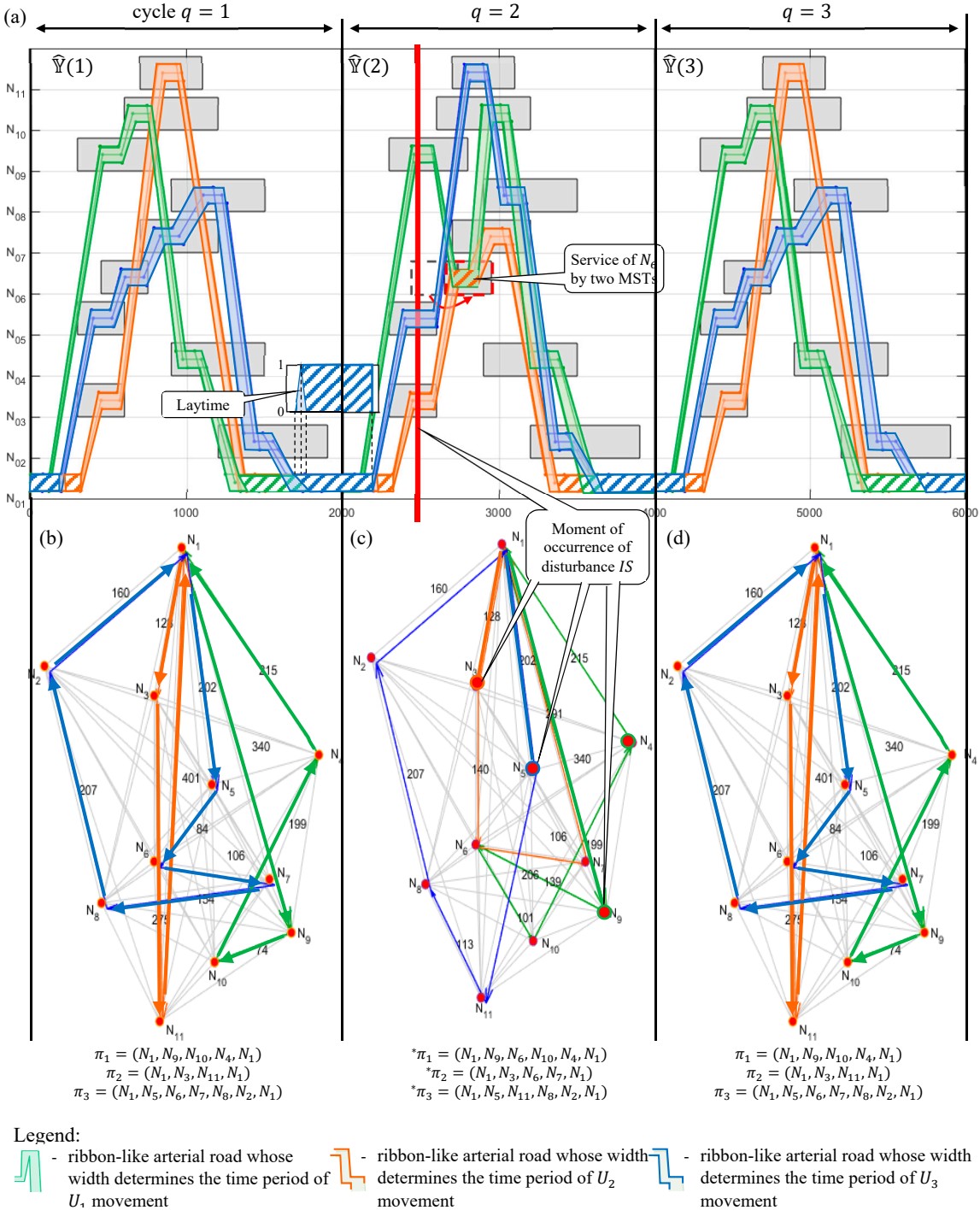

**Figure 7.** Cyclic fuzzy schedule (**a**); no disturbances (**b**); occurrence of the disturbance (**c**); and no disturbances (**d**).

In fuzzy schedule $^{*}\widehat{\mathbb{Y}}$ (Figure 7a), the operations are represented as ribbon-like "arterial roads", whose increasing width shows the time of vehicle movement resulting from the growing uncertainty. It is worth noting that the uncertainty is reduced at the end of each time window as a result of the operation of vehicles waiting at node $N_1$. The increasing uncertainty is not transferred to the subsequent cycles of the system. Uncertainty is reduced as a result of the implementation of OFN formalism. The MST waiting time at node $N_1$ has a negative orientation (laytimes $^{*}\widehat{w_1^1}$, $^{*}\widehat{w_1^2}$ and $^{*}\widehat{w_1^3}$). An example illustrating the use of standard fuzzy numbers for modeling the behavior of cyclic systems belonging to milk-run systems can be found in [44]. Taking the above into account, the proposed method of the dynamic planning of MSTs in cyclic maintenance delivery systems is unique, due to the possibility of taking into account the reduction in uncertainty in subsequent work cycles of the considered system.

Moreover, the routes $^{*}\pi_1$, $^{*}\pi_2$, $^{*}\pi_3$ remain unchanged (see routes $^{*}\pi_1$, $^{*}\pi_2$, $^{*}\pi_3$ in Figure 7a) until a disturbance occurs, and then they are rerouted, rescheduled and finally synchronized again so that all customers are serviced on time. This means that the model developed in this study allows to adjust the adopted delivery plans to disturbances changing the pre-established services timetable.

In addition to the above experiments, the effectiveness of the proposed approach was evaluated for the distribution networks of different sizes (different numbers of nodes and MSTs). The results are collected in Table 3.

**Table 3.** Results of the computational experiments carried out for the selected instances of distribution networks.

| Number of Nodes $n$ | Number of MSTs $K$ | Calculation Time (s) |
| :---: | :---: | :---: |
| 5 | 1 | <1 |
| 5 | 2 | <1 |
| 5 | 3 | <1 |
| 5 | 4 | <1 |
| 7 | 1 | <1 |
| 7 | 2 | <1 |
| 7 | 3 | 1 |
| 7 | 4 | 5 |
| 9 | 1 | 3 |
| 9 | 2 | 8 |
| 9 | 3 | 11 |
| 9 | 4 | 15 |
| 11 | 1 | 10 |
| 11 | 2 | 25 |
| 11 * | 3 | 31 |
| 11 | 4 | 67 |
| 13 | 1 | 22 |
| 13 | 2 | 61 |
| 13 | 3 | 108 |
| 13 | 4 | 124 |
| 15 | 1 | 46 |
| 15 | 2 | 115 |
| 15 | 3 | 215 |
| 15 | 4 | 380 |
| 17 | 1 | 250 |
| 17 | 2 | 554 |
| 17 | 3 | >900 |
| 17 | 4 | >900 |
| 20 | 1 | >900 |
| 20 | 2 | >900 |
| 20 | 3 | >900 |
| 20 | 4 | >900 |

*—the solution from Figure 7.

To summarize, the experiments were carried out for networks containing 5–20 nodes in which services were made by sets consisting of 1–4 MSTs (the sizes of the instances considered correspond to the sizes of the networks encountered in practice [45]). The aim of the experiments was to estimate the time necessary to designate the routes to guarantee timely services in the case of disturbances *IS* occurrence. In all instances considered, the synthesis of routes required considerable time expenditure. This means that the problems considered can be solved online mode when the size of the service distribution network does not exceed 15 nodes. In the case of larger networks, the effect of combinatorial explosion becomes of significant importance and limits the practical use of this method to the offline prototyping of possible variants of service mission scenarios.

## 7. Conclusions

The novelty of this study is that it proposed ordered fuzzy numbers algebra framework aimed at the solution of the DMRP, which was stated in terms of the fuzzy constraint satisfaction problem. The specificity of the process involved in the course of the maintenance delivery schedule planning results in the need to determine the sequentially cumulative uncertainty in the performance of the operations involved in it. In other words, the accumulation of uncertainties of previously performed operations result in the increasing uncertainty of the timely execution of subsequent operations. The question that arises in this context concerns the method for preventing additional uncertainty introduced by the combinations of summing up uncertainties of successively summed uncertain deadlines for the implementation of operations. In this context, in contrast to standard fuzzy numbers, the support of a fuzzy number obtained by algebraic operations performed on the ordered fuzzy numbers domain does not expand. In turn, however, the possibility of carrying out algebraic operations is limited to select domains of the computability of these supports. For this reason, sufficient conditions implying the calculability of arithmetic operations guarantee interpretability of the results obtained are proposed. Their use confirms the competitiveness of the analytical approach in relation to the time-consuming computer-simulation-based calculations of MST schedules.

The proposed framework enhanced by modern IT technology, e.g., Internet-of-Things, enables the digital integration of a vehicle fleet providing maintenance services to geographically dispersed customers, and provides feasible solutions forced by ad hoc emerging disturbances, i.e., delivering near-optimal schedules prioritizing the just-in-time performance of maintenance services and the execution of a maximum of the many orders among those reported during the mission. The results of the conducted tests demonstrate that the proposed analytical approach enables to cope with the problems of dynamic routing and scheduling of mobile teams servicing customer requests while taking into account the uncertainty of the travel time and provided maintenance times. In this sense, the paper presents the method enabling to generate alternative MST routing scenarios to customer request change. Its implementation in DSS will support decision-making activities undertaken by service MSTs dispatcher.

The results of the conducted experiments indicate the implementation of the relevant methods in systems supporting the reactive scheduling of MSTs following the milk-run driven manner. In this context, the use of available environments, such as IBM ILOG CPLEX, ECLiPSe, Gurobi, etc., which make it possible to tackle the practical-scale problems, can be viewed as an attractive solution for problem-oriented DSS. It is also worth noting that the research conducted, being in line with the concept of Maintenance 4.0 which stresses the need to seek solutions that allow information systems to create a virtual copy of the physical world, and provides a programming framework for context-aware information model design.

In future work, some additional factors including the impatient customer concept [46], refilling stops, and synchronization of works carried out for a given user by various service teams, will be recorded and streamlined into the proposed approach. Furthermore, the currently studied problem will be extended to the dynamic planning of multi-period outbound MST-driven services, delivery aimed at scheduling being implemented in a rolling horizon approach [47].

**Author Contributions:** Z.B. and M.J.-K. prepared the state of art, G.B. formulated the problem and proposed the model, G.B. and P.N. conceived and designed the experiments; Z.B. and G.B. prepared the dynamic algorithm; M.J.-K. analyzed the data; G.B., M.J.-K., P.N. and Z.B. wrote the paper. All authors have read and agreed to the published version of the manuscript.

**Funding:** This research received no external funding.

**Acknowledgments:** This research was carried out under the internship: Declarative models of the vehicles fleet mission planning (Aalborg University, 23 November 2020–31 March 2021).

**Conflicts of Interest:** The authors declare no conflict of interest.

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
