# Peer review of "Dynamic Planning of Mobile Service Teams’ Mission Subject to Orders Uncertainty Constraints"

_applsci, doi:10.3390/app10248872_

Round 1
Reviewer 1 Report
The manuscript is focused on the problem of optimising cyclical deliveries by more than one vehicle. The authors consider a case with disturbances in the form of unknown travel times between delivery points, fluctuations in duration of loading/unloading operation. As a solution to this problem, an approach using fuzzy numbers transformed into OFN algebra and programming with constraints was proposed.
The article lacks reference to other methods of processing uncertain data. Why was OFN algebra chosen and not, for example, probability distributions? Because OFN algebra is not widespread in the scientific world, this aspect should be better substantiated.
The disadvantage of this manuscript is the lack of comparison of the proposed method with other methods known from the literature. It should be taken into account that the problem formulated by the authors has been present in the literature for a long time and the authors should indicate where their solution exceeds the previous one. It is not enough to present only a different calculation method; a tangible gain from its introduction should be presented, e.g. better schedules, shorter calculation time compared to the methods already known.
Because we are dealing with random events, how should service deadlines be understood? Are they rigid? I would expect that in this type of model, we can expect delivery to be done with a certain probability, which should not be lower than a predetermined value. What are the consequences of late delivery in the model?
A change in the delivery point for MSTs as a result of a disruption is only possible for a delivery, such a change for collecting a parcel is not possible. Does the model take this factor into account? The manuscript should describe rerouting options in more detail.
The fact that processing time very quickly achieves unacceptable values is also cause for concern. If we take into account the fact that the proposed solution might be used for parcels delivery even in a medium-sized city, the number of delivery points (nodes) would reach hundreds and the number of couriers (MST) would be more than ten, then unfortunately, this solution will not be suitable.
Minor comments:
- Abbreviation OFN should be introduced.
- Notation is not consistent. For example, there are sk and sk. Are they the same or different values?
Author Response
The authors would like to thank the referees for the very helpful, kind, and careful comments and suggestions. The authors have incorporated all the referees’ comments in the revision of the paper and have done the following:
R1. The article lacks reference to other methods of processing uncertain data. Why was OFN algebra chosen and not, for example, probability distributions? Because OFN algebra is not widespread in the scientific world, this aspect should be better substantiated.
Response: Thank you very much for your question allowing us to describe how we take the "human factor" influence on the VRP into consideration.
This is a result of the very nature of MSTs, in which the time of carrying out the operations of transport and periodic inspections/repairs depends on both the transport infrastructure and the prevailing weather conditions as well as on human factors. The imprecise nature of these parameters is implied due to the operator's psychophysical disposition changing over time, the scale of performed repairs, time-consuming tests and measurements, means of transport used, disturbances in the flow of traffic (e.g. traffic lights failures, road accidents, etc.). Therefore, the time values of the operations performed vary and are uncertain. Consequently, it is also impossible to precisely define the times of MSTs arrival to customers ordering maintenance services (i.e. the delivery schedule is not precise). The non-stationary nature of the uncertainty of the parameters mentioned, and the usually small set of available historical samples in practice limits the choice of a formal data model to a fuzzy numbers driven one. The specificity of the process involved in the course of the maintenance delivery schedule planning, results in the need to determine the sequentially cumulative uncertainty in the performance of the operations involved in it. In other words, the accumulating uncertainty of previously performed operations result in increasing uncertainty of timely execution of subsequent operations. The question that arises in this context concerns the method for avoiding additional uncertainty introduced in the combinations of summing up uncertainties of successively summed uncertain deadlines for the implementation of operations. In this context, in contrast to standard fuzzy numbers, the support of a fuzzy number obtained by algebraic operations performed on the ordered fuzzy numbers domain does not expand. In turn, however, the possibility of carrying out algebraic operations is limited to selected domains of the computability of these supports. That is a reason why we propose sufficient conditions implying the calculability of arithmetic operations that guarantee the interpretability of the results obtained. They use confirms the competitiveness of the analytical approach in relation to time-consuming computer-simulation-based calculations of MST schedules. The following paragraph has been added: Page 14. The specificity of process involved in the course of the maintenance delivery schedule planning results in the need to determine the sequentially cumulative uncertainty in the performance of the operations involved in it. In other words, the accumulation of uncertainties of previously performed operations result in increasing uncertainty of timely execution of subsequent operations. The question that arises in this context concerns the method for avoiding additional uncertainty introduced by the combinations of summing up uncertainties of successively summed uncertain deadlines for the implementation of operations. In this context, in contrast to standard fuzzy numbers, the support of a fuzzy number obtained by algebraic operations performed on the ordered fuzzy numbers domain does not expand. In turn, however, the possibility of carrying out algebraic operations is limited to selected domains of the computability of these supports. For this reason sufficient conditions implying the calculability of arithmetic operations guarantee interpretability of the results obtained are proposed. Their use confirms the competitiveness of the analytical approach in relation to time-consuming computer-simulation-based calculations of MST schedules. R2. The disadvantage of this manuscript is the lack of comparison of the proposed method with other methods known from the literature. It should be taken into account that the problem formulated by the authors has been present in the literature for a long time and the authors should indicate where their solution exceeds the previous one. It is not enough to present only a different calculation method; a tangible gain from its introduction should be presented, e.g. better schedules, shorter calculation time compared to the methods already known.
Response: Thank you very much. Referring to your comment, it is worth noting that the naturally achieved reduction of uncertainty in cyclic systems as a result of planned standstills on buffers, after stoppages in replenishment bases, breaks after the performance of services and in continuous driving, and so on, has so far not been possible to model. Models and methods offered by Fuzzy Sets and/or Probability theory turned out to be ineffective. This is due to in both cases, algebraic operations on fuzzy/random variables result in increasing uncertainty. In particular it means, that the use of standard fuzzy numbers leads to an additional increase in the uncertainty of the results of arithmetic operations carried out on them. It is worth noting that this observation confirms the Zadeh’s extension principle, the uncertainty of variables increases with successive cycles of the system operation, until the information about their value is no longer useful. Relevant studies confirming this fact were carried out on an example illustrating the use of standard fuzzy numbers for modeling the behavior of cyclic systems belonging to milk-run systems (Bocewicz et al. 2020). Taking the above into account, the proposed method of dynamic planning of MST in cyclic maintenance delivery systems seems to be unique, due to the possibility of taking into account the reduction of uncertainty in subsequent work cycles of the considered system.
The following paragraph has been added:
Page 13. An example illustrating the use of standard fuzzy numbers for modeling the behavior of cyclic systems belonging to milk-run systems can be found in [44]. Taking the above into account, the proposed method of dynamic planning of MST in cyclic maintenance delivery systems is unique, due to the possibility of taking into account the reduction of uncertainty in subsequent work cycles of the considered system.
Has been added to the references list:
[44] Bocewicz G., Banaszak Z., Rudnik K., Witczak M., Smutnicki C., Wikarek J., Milk-run routing and scheduling subject to fuzzy pickup and delivery time constraints: An ordered fuzzy numbers approach. IEEE International Conference on Fuzzy Systems, 2020, art. no. 9177733, doi:10.1109/FUZZ48607.2020.9177733
R3. Because we are dealing with random events, how should service deadlines be understood? Are they rigid? I would expect that in this type of model, we can expect delivery to be done with a certain probability, which should not be lower than a predetermined value. What are the consequences of late delivery in the model?
Response: Thank you very much. For clarification, note, that uncertainty of service operations assumed in this study is represented by an interval . Consequently, the solutions (in the form of MST routes) that will guarantee the service of each customer in a given time , regardless of the variability of traveling times between nodes (see example Fig. 5) are sought. This approach avoids the costly probability calculation.
R4. A change in the delivery point for MSTs as a result of a disruption is only possible for a delivery, such a change for collecting a parcel is not possible. Does the model take this factor into account? The manuscript should describe rerouting options in more detail.
Response: Thank you very much. For clarification, note that a disruption of changing the delivery point for MSTs is not considered. It was assumed that only the service deadline can be changed. The consequence of the adopted assumption is the possibility of assessing the permissible changes to the pre-planned routes (i.e., reroutings) that reflect the changes caused by newly introduced service deadlines. Disruption also taking into account the change of customers’ location of use will be the subject of future work. So, thank you very much for so valuable suggestion.
R5. The fact that processing time very quickly achieves unacceptable values is also cause for concern. If we take into account the fact that the proposed solution might be used for parcels delivery even in a medium-sized city, the number of delivery points (nodes) would reach hundreds and the number of couriers (MST) would be more than ten, then unfortunately, this solution will not be suitable.
Response: Thank you very much. The lack of an acceptable solution means that the customer, who incurred an unforeseen change in service deadline, cannot be covered by the available fleet of vehicles. In such situations, it is possible to use reserve vehicles (prepared for emergencies) or omit the service (lower priority) clients. Both approaches will be the subject of future work. So, once again thank you very much for so valuable suggestions.
R6. Minor comments: 1. Abbreviation OFN should be introduced. 2. Notation is not consistent. For example, there are sk and sk. Are they the same or different values?
Response: Thank you very much. The mentioned mistakes have been corrected.

Reviewer 2 Report
The authors aim to propose a solution procedure to solve the dynamic routing problem. The situation of the illustrative example should be more realistic to provide a proof of their contribution. The obtained results can not be traced.
Author Response
The authors would like to thank the referees for the very helpful, kind, and careful comments and suggestions. The authors have incorporated all the referees’ comments in the revision of the paper and have done the following:
R1. The authors aim to propose a solution procedure to solve the dynamic routing problem. The situation of the illustrative example should be more realistic to provide a proof of their contribution. The obtained results can not be traced.
Response: Thank you very much. For clarification, note that the example under consideration addresses many different real-life problems, in particular regarding recurring (cyclical) customer service distributed networks. These types of problems occur in situations related to: delivering ready-made meals to order (catering services), garbage collection, service of water stations, mobile medical care (serving chronically (terminally) ill patients), mail order sale (e-commerce) especially in the time of Covid-19 crisis, etc. In the presented situations, disturbances often occur, caused by changes of the delivery date, i.e. covering the change of dates of ordered services (change the date of the ordered meal, the date of garbage collection, the date of the water supply infrastructure inspection, etc.).

Reviewer 3 Report
The paper presents a work of excellent scientific quality and very well structured and described. A complete literature review is carried out and the use of a very adequate methodology for the problem is proposed.
The mathematical model is very well described but I recommend to the authors some minor corrections or clarifications:
Line 72, page 2: missing "k" in "mil-run"
Line 103 page 3: repetitive sentence
Line 120 page 3: missing "," after "...mission carried out"
Line 228 page 5: extra blank space after "numbers"
Line 235 page 5: the acronym OFN should be previously defined
Line 274 page 6: what represents variable t_i?
Line 282 page 7: I think that te correct interval time is [900,1500]
Line 559 page 17: missing table number in reference to Table 3.
Finally, congratulations to the authors for the work.
Author Response
The authors would like to thank the referees for the very helpful, kind, and careful comments and suggestions. The authors have incorporated all the referees’ comments in the revision of the paper and have done the following:
Reviewer #3
R1. The mathematical model is very well described but I recommend to the authors some minor corrections or clarifications:
Line 72, page 2: missing "k" in "mil-run"
Line 103 page 3: repetitive sentence
Line 120 page 3: missing "," after "...mission carried out"
Line 228 page 5: extra blank space after "numbers"
Line 235 page 5: the acronym OFN should be previously defined
Line 274 page 6: what represents variable t_i?
Line 282 page 7: I think that te correct interval time is [900,1500]
Line 559 page 17: missing table number in reference to Table 3.
Finally, congratulations to the authors for the work.
Response: Thank you very much. The mentioned mistakes have been corrected.
